# Cyberbullying Among Adolescents in Norway: Time Trends and Factors Associated with Perpetration and Victimization

**DOI:** 10.3390/bs14111043

**Published:** 2024-11-05

**Authors:** Tore Bonsaksen, Annette Løvheim Kleppang, Anne Mari Steigen

**Affiliations:** 1Department of Health and Nursing Science, Faculty of Social and Health Science, Inland Norway University of Applied Sciences, 2406 Elverum, Norway; anne.steigen@inn.no; 2Department of Health, Faculty of Health Sciences, VID Specialized University, 4024 Stavanger, Norway; 3Department of Public Health and Sports Sciences, Inland Norway University of Applied Sciences, 2406 Elverum, Norway; annette.kleppang@inn.no

**Keywords:** adolescents, cyberbullying, depression, loneliness, mental health, Norway

## Abstract

There is limited knowledge about time trends in cyberbullying involvement, and whereas previous studies have often focused on various mental health outcomes, possible outcomes of cyberbullying may concern a wide range of areas. A better understanding of cyberbullying development over time, as well as of the possible consequences, is needed for effective prevention and intervention efforts. The aims of this study were (i) to gain knowledge of time trends in cyberbullying among Norwegian adolescents, and (ii) to better understand how different forms of cyberbullying involvement is related to a variety of outcomes. Data from the Norwegian Ungdata surveys (2014–2016) were used to assess involvement in cyberbullying over time. Associations with cyberbullying involvement and four outcome variables (belief in a good and happy life, loneliness, school thriving, and depressive symptoms) were assessed using logistic regression analyses. Across years of study, the prevalence rates for cyberbullying involvement were 1.5% for perpetration, 3% for victimization, and 1% for perpetration-victimization. Compared with non-involvement, involvement in cyberbullying in any role was associated with poorer outcomes on all variables. Cyber-victims who were also perpetrators were less likely to have high levels of loneliness and depressive symptoms, compared to cyber-victims only. The prevalence of cyberbullying involvement among adolescents in Norway was relatively unchanged between 2014 and 2016. Cyberbullying involvement is associated with negative outcomes related to mental health, loneliness, school thriving, and beliefs in one’s personal future, and victims of cyberbullying appear to be the most vulnerable group.

## 1. Introduction

Adolescents spend increasingly more time on the internet and social media. In Norway, adolescents spent an average of 4–6 h online daily in 2022, representing a doubling since 2015 [1]. Consequently, social interaction and communication between adolescents take place on digital platforms to a large and increasing degree. The developing world of electronic social media technology and its widespread integration into the daily routines of adolescents create alternative social environments in which young people engage in relationships [2]. This development implies that negative online social interactions, such as cyberbullying, have become new threats to adolescents’ mental health and well-being [3,4]. Adolescents in the middle age range appear to be more at risk of experiencing cyberbullying victimization, while younger children and adolescents approaching adulthood have been found to be less exposed. For example, Williams and Guerra [5] found that 4.5% of fifth-graders, 12.9% of eight-graders, and 9.9% of high school students had been victims of cyberbullying, indicating a possible curvilinear relationship between cyberbullying exposure and age. While traditional face-to-face bullying still appears to be more prevalent than cyberbullying [6,7], studies have found substantial overlap between the two forms of bullying [8,9,10,11,12]. Exposure to both forms—and particularly in combination—has been associated with higher levels of mental health problems such as depression, anxiety, loneliness, suicidal behavior, and self-harm [9,13,14,15] and lower levels of subjective well-being and life satisfaction [16,17].

In this study, we build on Hinduja and Patchin [18] (p. 11) who defined cyberbullying as the “willful and repeated harm inflicted through the use of computers, cell phones and other electronic devices”. Thus, cyberbullying includes bullying that occurs via e-mail, in chatrooms, via instant or text messaging, and on websites, blogs, and social media [19]. However, different definitions of cyberbullying exist in the literature, and there is no consensus on one common definition. Concepts of cyberbullying also differ—while some suggest that cyberbullying is largely an online extension of face-to-face bullying, others claim that cyberbullying differs substantially from traditional bullying [20]. For example, the repetitive aspect of bullying may be different online. A hurtful message or photo may be posted once, and then the harm is repeated through the sharing by others. Photos and posts may also be difficult to remove from digital platforms, rendering even one-time victims of cyberbullying with a relentless burden [21]. Cyberbullying is also characterized by anonymity [22,23]; hence, victims may not know who the bully is and perpetrators may feel safe from scrutiny and criticism [24]. Moreover, the physical distance between the bully and the victim is unique for cyberbullying, and the bully’s inability to see the victim’s reaction may increase the risk of perpetration [25,26].

Studies and reviews have reported widely varying prevalence rates of different types of cyberbullying involvement, e.g., [10,27,28,29,30,31,32,33]. One review found that prevalence rates ranged between 1% and 61% for cyber-victims, between 3% and 39% for perpetrators, and between 2% and 72% for cyberbully-victims [34]. A challenge for cyberbullying research is the lack of standardized measurement and definitions of time periods, and this can make it difficult to compare prevalence rates across studies. However, given the rapid increase in the use of social media and online interactions between adolescents during the last decade [1], increased rates of cyberbullying involvement may be expected. A Norwegian study using data from 2015 found one-month prevalence rates of 3% for cyber-victims, 1% for perpetrators, and 1% for cyberbully-victims [14], whereas data from a survey conducted in Norwegian schools in 2022 found that 2.7% of students aged 10–19 had been cyberbullied twice or more often during the last month [35]. While these findings are certainly in the lower range, compared with international research findings, they may reflect the state of cyberbullying involvement at specified time points among adolescents in Norway.

In Norway, Skilbred-Fjeld and co-workers [14] found that adolescents involved in cyberbullying either as victims, perpetrators or cyberbully-victims were significantly more often involved in self-harm, suicide attempts, and antisocial behavior, and had higher risk of anxiety and depression, compared to adolescents not involved in cyberbullying. When comparing the three groups involved in cyberbullying, the only significant difference was the lower proportion of perpetrators reporting suicide attempts compared to cyber-victims and cyberbully-victims. Thus, involvement in cyberbullying in any role—victim, perpetrator, or both—appears to be associated with poorer psychosocial health, while differences between perpetrators, cyber-victims, and cyberbully-victims appear to be less outspoken. However, comparing outcomes among those involved in different cyberbullying roles appears to be a relatively new line of research which warrants more attention.

There is limited research on time trends in cyberbullying. Some panel studies on bullying incorporating multiple waves of data collection spanning several years have included cyberbullying in the last wave only, with no opportunity to compare prevalence rates across waves [36]. However, one recent US American review reported time trends in cyberbullying between 2009 and 2022 from a race and ethnicity perspective. For all groups, except for Asian high school girls, an increase over time was reported [37]. Another study of Australian adolescents found that cyber victimization increased between 2015 and 2020, while cyber perpetration did not [38]. Although unrelated to concurrent increases in mental health problems, they also found a sharp rise in both cyber victimization and cyber perpetration during the early stages of the COVID-19 pandemic. Thus, it appears that a greater reliance on internet-based means of communication and socialization can give rise to increased cyberbullying involvement. However, most longitudinal studies on cyberbullying have been found to utilize data from a short period of time, and the need for greater time-lag between the waves of data collection has been highlighted [39]. The nature of the relationships between cyberbullying and other factors also needs more research attention. Thus far, a meta-analysis of 56 studies has shown that probable causes and effects of cyberbullying perpetration and victimization were reciprocal [39]. Cyberbullying perpetration predicted subsequent externalizing problem behaviors, but problem behaviors and higher levels of internet use also predicted perpetration. Conversely, cyberbullying victimization predicted internalizing problems like anxiety and depression over time, but anxiety, depression, and internet use also predicted subsequent victimization.

In summary, while the use of social media and digital communication technologies has become an integral aspect of adolescents’ daily lives, there is still limited knowledge about time trends in cyberbullying involvement. Moreover, possible outcomes related to cyberbullying may concern a wide range of areas, whereas previous studies have often focused on various mental health outcomes. There is a need for studies that can document the development in cyberbullying, and to assess the range and severity of consequences it may have for those involved. To the authors’ knowledge, such studies have not been previously conducted in Norway. Therefore, the aims of this study were (i) to gain knowledge of the time trends in cyberbullying among Norwegian adolescents, and (ii) to better understand how different forms of cyberbullying involvement are related to a variety of outcomes. Specifically, based on previous studies, the hypotheses were that: (i) cyberbullying involvement would increase between 2014 and 2016, and (ii) cyberbullying involvement would be associated with higher levels of depressive symptoms and loneliness, and lower levels of school thriving and positive belief in one’s future.

## 2. Materials and Methods

### 2.1. Data Sources and Participants

We employed data from the Ungdata surveys conducted between 2014 and 2016. The Ungdata surveys are surveys of Norwegian adolescents aged between 13 and 19 years. They are conducted annually among adolescents in Norwegian junior and senior high schools and provide information about adolescents’ health, well-being, attitudes, and behaviors (www.ungdata.no; accessed on 1 September 2024). All adolescents in the relevant age range who are students in one of the participating schools are invited to participate. There are no exclusion criteria. Each municipality is recommended to participate every third year. Norwegian Social Research (NOVA) at Oslo Metropolitan University is responsible for the survey in collaboration with the Regional Drug and Alcohol Competence Centers (KORUS).

### 2.2. Measures

*Cyberbullying* perpetration was assessed with the following question: “Do you sometimes participate in bullying or threats against other adolescents via internet or mobile phone?”. Cyberbullying victimization was assessed with the following question: “Have you been exposed to bullying or threats from other adolescents via internet or mobile phone?”. Both questions had the response options ‘yes, several times a week’ (1), ‘yes, approximately once a week’ (2), ‘yes, approximately once every two weeks’ (3), ‘yes, approximately once a month’ (4), ‘almost never’ (5), and ‘never’ (6) [40]. In line with previous studies, e.g., [41,42] and with a conceptualization of bullying as “something that occurs over time and with a repetitive nature” [43], a cut-off between perpetrators/non-perpetrators and between victims/non-victims was set between (3) and (4). In effect, cyberbullying perpetration was defined as having exposed other adolescents to cyberbullying once every two weeks or more often, whereas victimization of cyberbullying was defined as having been exposed once every two weeks or more often. Based on the above, the participants’ cyberbullying status was defined as not involved in cyberbullying as perpetrator nor as victim (reference), as perpetrator but not victim (1), as victim but not perpetrator (2), or as both perpetrator and victim (3). In the years after 2016, cyberbullying was assessed with questions that do not align well with the questions used before and after that time. Consequently, we did not assess cyberbullying involvement after 2016.

*Belief in a good and happy life* was measured with one item: “What do you think your future will be like: Do you think that you will have a good, happy life?”. Response options were ‘yes’ (1), ‘no’ (2), and ‘don’t know’ (3) [40]. In the analysis, we distinguished between participants who responded affirmatively (1) and those who disconfirmed or were unsure (0).

*Loneliness* was measured with one item. Its opening phrase was “During the past week, have you been affected by any of the following issues”, and then stated, “felt lonely”. Response options were ‘not been affected at all’ (1), ‘not been affected much’ (2), ‘been affected quite a lot’ (3), and ‘been affected a great deal’ (4) [40]. A cut-off was applied to distinguish between adolescents with lower (1 and 2) and higher (3 and 4) levels of loneliness.

*School thriving* was measured with one item. First, a statement prepared the adolescents for several questions concerning the situation at school: “Do you agree or disagree with the following statements about your situation at school?”. Then, the specific item was phrased “I enjoy school”. Response options were ‘totally agree’ (1), ‘somewhat agree’ (2), ‘somewhat disagree’ (3), and ‘totally disagree’ (4) [40]. The cut-off was set to distinguish between those who agreed with the statement (1 and 2) and those who did not (3 and 4).

*Depressive symptoms* were assessed with a six-item scale derived from the Depressive Mood Inventory [44], in turn based on the Hopkins Symptom Checklist [45]. The adolescents indicated the degree to which they had been affected by the following during the past week: ‘Felt that everything is a struggle’ (item 1), ‘had sleep problems’ (item 2), ‘felt unhappy, sad or depressed’ (item 3), ‘felt hopelessness about the future’ (item 4), ‘felt stiff or tense’ (item 5), and ‘worried too much about things’ (item 6). Each item had four response categories: ‘not been affected at all’ (1), ‘not been affected much’ (2), ‘been affected quite a lot’ (3), and ‘been affected a great deal’ (4). Sum scores were computed, ranging from 6 to 24, with higher scores indicating higher depressive symptom levels. In this study, a cut-off was established at the 80th percentile (i.e., symptom score of 18 or higher), distinguishing between participants with depressive symptom ratings in the 20% highest range and those with lower ratings. The scale has been psychometrically evaluated among Norwegian adolescents in previous Ungdata-based studies, demonstrating good reliability (Person Separation Index: 0.802) and appearing to work reasonably well at an overall level [46].

*Friendship* was measured with one item: “Do you have at least one friend who you trust completely and who you can tell absolutely anything?”. Response options were ‘yes, definitely’ (1), ‘yes, I think so’ (2), ‘I don’t think so’ (3), and ‘there is nobody I would call a friend at the moment’ (4). For the analysis of this study, we distinguished between those who reported ‘yes, definitely’ (1) and those who reported otherwise (0). School grade was used as an indicator for *age,* while *gender* was reported as male or female. *Socioeconomic status* was measured with three items derived from the family affluence scale (FAS) [47,48]. The first question was ‘Does your family have a car?’, with response options ‘no’, ‘yes, one’, and ‘yes, two or more’. The second was ‘Do you have your own bedroom?’, with response options ‘yes’ or ‘no’. The third was ‘How many times have you traveled somewhere on holiday with your family over the past year?’, with response options ‘never’, ‘once’, ‘twice’, and ‘more than twice’. All items were coded so that higher scores indicated higher levels of family affluence, with scale scores ranging between 3 (lowest affluence) and 9 (highest affluence). A fourth FAS question was added to the Ungdata surveys beginning in 2017; therefore, it was not included in the current study using data from before that time.

### 2.3. Data Analysis

Participants with missing data were removed from the analyses casewise (analysis by analysis), with the exception of the multivariate analyses which required no missing data on the employed variables. Thus, sample size varied between analyses, and the actual included *n* is reported for each analysis. Descriptive analyses of the sample were included for age, gender and socioeconomic status by year of data collection. To examine bivariate relationships with cyberbullying status, this variable was cross tabulated with all outcomes: belief in a good and happy life, loneliness, school thriving, and depressive symptoms, and differences in proportions between groups were tested with Chi Square tests of independence. To obtain measures of adjusted associations with cyberbullying status, multiple logistic regression analyses were used. Cyberbullying status (i.e., not involved in cyberbullying, perpetrator, victim, both perpetrator and victim) was the predictor variable, the analysis was performed for each of the four outcomes separately, and adjustment was made for age, gender, and friendship in each analysis. Control variables were selected based on their known relationships with the outcome variables, so that detected associations between cyberbullying and the outcomes would not be confounded by these influences. Odds ratio (OR) was provided as effect size along with its corresponding 95% confidence interval (CI). Statistical significance was set at *p* < 0.05.

### 2.4. Research Ethics

The Ungdata studies were conducted in accordance with the Declaration of Helsinki [49]. Informed consent was received through passive consent from all adolescents and their parents (if the adolescent was a minor). Prior to the adolescents’ participation, adolescents and their parents were informed about the purpose and procedures of the Ungdata study and were informed that they could withdraw from the study at any time. The information letter was approved by the Norwegian Centre for Shared Services in Education and Research. At the time of receiving the web-based questionnaires in school, the adolescents decided whether they wanted to participate or not. They were informed that participation was voluntary and that any question could be skipped. A teacher or administrator was present to answer any questions from the adolescents during the data collection. It took about 30–45 min to complete the questionnaire.

All data were collected anonymously and then analyzed by independent researchers who did not participate in the collection of the data. The current study was approved by the Research Ethics Committee at Inland Norway University of Applied Sciences (protocol code: 23/05377).

## 3. Results

### 3.1. Participation Across the Study Period

In all study years, junior high school students had a higher representation than senior high school students, while the gender composition was close to 50–50 during the period. The age and gender of the participants across the study period are shown in Table 1.

### 3.2. Cyberbullies and Victims of Cyberbullying

Reports of being engaged in online bullying or bullying via mobile phone decreased slightly from 1.6% in 2014 to 1.2% in 2016. Victimization of online bullying or bullying via mobile phone was at or slightly below 3% across the period, while the proportion of perpetrators who were also victims was 0.7–0.8% each year. The results are shown in Figure 1.

### 3.3. Cyberbullying Status in Relation to Outcomes

Cyberbullying status (i.e., neither perpetrator nor victim, perpetrator but not victim, victim but not perpetrator, both perpetrator and victim) was cross tabulated against several outcomes: optimistic beliefs in one’s personal future, loneliness, thriving at school, and depressive symptoms. The results for all years together are shown in Table 2. Participants who were neither perpetrators nor victims reported the best prospects for a good and happy life, and they were less lonely, reported more thriving in school, and were less inclined to have a high level of depressive symptoms, compared to their counterparts in the other groups. Across outcomes, perpetrators who were not themselves victims were better off than those who were victims, but not perpetrators themselves. Participants who were victims, but not perpetrators, were similar to those who were both perpetrators and victims with regard to their belief that they would live a good and happy life, and with regard to thriving in school. However, victims who were not perpetrators themselves were considerably more often lonely and had a high level of depressive symptoms, compared to those who were victims and also perpetrators. The year-specific analyses showed an identical pattern within each year, with only minor variations.

Adjusted measures of associations between cyberbullying status and the outcomes were produced by a series of logistic regression analyses, using optimistic belief in a good and happy life, loneliness, school thriving, and depressive symptoms as subsequent dependent variables. Lower age, male gender, and having at least one friend were associated with higher likelihood of foreseeing a good and happy life and of thriving at school, and lower likelihood of being lonely and having a high level of depressive symptoms. Being a cyberbullying perpetrator or victim was associated with lower likelihood of foreseeing a good and happy life and with thriving in school, and with higher likelihood of being lonely and having a high level of depressive symptoms. Those who were victims and also perpetrators were lonely less often and had a high level of depressive symptoms less often, compared to those who were victims but not perpetrators. The results are shown in Table 3.

## 4. Discussion

### 4.1. Prevalence of Cyberbullying Involvement

The prevalence rates for cyberbullying perpetration (about 1.5%), victimization (about 3%) and the combination of both perpetration and victimization (slightly below 1%) were similar in the study period. These results are strikingly similar to the results reported in a previous Norwegian study using data from 2015 [14], supporting the notion that the prevalence of cyberbullying perpetration and victimization among Norwegian adolescents has been relatively stable between 2014 and 2016, and that it seems to be at comparably low levels considering the results reported in international studies, e.g., [10,27,28,29,30,31,34]. However, our lack of access to comparable data from 2017 onwards is unfortunate, and further research is needed to show the subsequent development in cyberbullying involvement among adolescents in Norway.

### 4.2. Cyberbullying Involvement and Associated Outcomes

The results demonstrated that involvement in cyberbullying—either as perpetrator, victim, or both—was associated with poorer outcomes in all defined areas. These results are well aligned with previous studies, demonstrating that both cyberbullying and cybervictimization are associated with higher levels of mental health problems in a range of areas [9,13,14,30,50,51,52,53], as well as higher levels of problems with attention, emotion regulation, relations with peers, unauthorized school absence, and other social problems [14,32,53,54,55]. Moreover, results from a previous study showed that online and offline victimization was positively linked to depression, and subsequently to an increased likelihood of engaging in cyberbullying perpetration [56]. Thus, involvement in cyberbullying in any role is associated with poorer outcomes across several areas, compared to non-involvement.

In addition, we found intriguing differences between victims of cyberbullying, perpetrators, and cyberbully-victims. The results showed that adolescents who were both perpetrators and victims were lonely considerably less often and had a high level of depressive symptoms less often, compared to adolescents who were victims but not perpetrators. While prior research has examined outcomes according to cyberbullying roles, the patterns revealed in these studies have not been easily interpretable. For example, a study from Portugal found that cyberbully-victims used more alcohol compared to cyber-victims only, and that cyber-victims liked school more than perpetrators, while no differences were revealed in other areas [57]. In Norway, perpetrators were shown to have lower risk of suicide attempts compared to cyber-victims and cyberbully-victims, whereas no differences were shown in other areas [14].

The current study’s results support the notion that among victims of cyberbullying, involvement in cyberbullying perpetration may—at least in the short term—buffer against or relieve depressive symptoms and loneliness. Potentially, victims of cyberbullying might find some relief by engaging in externalizing behaviors themselves, such as substance use, stealing, and acts of aggression, as suggested in one previous review [15]. Thus, victimization experiences should be considered a risk factor for cyberbullying perpetration [29,56,58]. Perhaps paradoxically, involvement in cyberbullying perpetration might also create a sense of belonging with others, which might link directly with lower levels of loneliness. This is in line with a study conducted during the COVID-19 pandemic, where Pfetsch and co-workers [59] found that cyberbullying was associated with higher levels of well-being, and, in particular, among adolescents with a high need to belong. With regard to belief in a good and happy life and school thriving, differences between these categories of involvement in cyberbullying were absent or marginal. Thus, given the clear differences in loneliness and depressive symptoms, it appears that such differences are only viable in some areas and not others.

### 4.3. Study Strengths and Limitations

This study is based on large samples of Norwegian adolescents, which is a major strength of the study. Our use of data from three subsequent years is also a strength, while future studies are still needed to show the development of cyberbullying involvement from 2017 onwards. Unfortunately, data on cyberbullying perpetration were not available after 2020, and the questions and response options used to assess cyberbullying perpetration and victimization during 2017–2019 were considered incomparable to the ones used during the other time periods. In view of these limitations, and the uncertain validity of the measuring method, the results should be interpreted with caution.

While using data from several periods in time allowed for examining time trends, the lack of longitudinal data from the same individuals precludes us from examining the temporal associations between cyberbullying involvement and the relevant outcomes. Therefore, like other studies based on analyses of cross-sectional associations, we cannot conclude on directionality. While there is reason to assume negative mental health effects of exposure to cyberbullying, studies have also found poor mental health to predict subsequent cyberbullying victimization [60] or have found effects to be cyclical and self-strengthening [39]. Furthermore, the associations found between cyberbullying status and outcomes might be influenced by other factors that were not controlled for in the present study, and the use of self-reported measures may have led to unidentified misclassification or measurement errors.

## 5. Conclusions

Despite the rapid increase in the use of online communication among adolescents during the last decade, our results suggest that the prevalence of cyberbullying perpetration and victimization, as well as the prevalence of cyberbully-victims, have been stable and largely unchanged between 2014 and 2016. However, one should consider that comparable data were missing from 2017 onwards.

Compared with adolescents not involved in cyberbullying, involvement in any cyberbullying role (cyberbully, cyber-victim, or both) was associated with poorer outcomes in all areas. Among adolescents involved in cyberbullying, perpetrators were less inclined than victims to have high levels of loneliness and depressive symptoms. Moreover, cyberbully-victims were less inclined than victims to have high levels of loneliness and depressive symptoms.

These results indicate that adolescents that in some way are involved in cyberbullying—as perpetrators, victims, or both—may need help to alleviate the burden of depressive symptoms and loneliness. While victims suffer the worst mental health consequences and therefore should be given particular attention among teachers and health and social care professionals, the risk of depressive symptoms and loneliness among cyberbullying perpetrators was also considerably higher than for those not involved. Thus, when designing policies and interventions targeting adolescent perpetrators, one should consider that these may struggle with a substantial mental health burden themselves. The results also indicate that maladaptive and aggressive coping attempts among victims may be considered as providing some emotional relief among the victims themselves, as they can potentially reduce the burden of depressive symptoms and feelings of loneliness. Thus, policies and interventions targeting adolescent victims of cyberbullying should also consider the potential risk among cyber-victims for becoming perpetrators themselves.

Further research is needed to document the more recent development in cyberbullying involvement among adolescents in Norway. While concurrent associations with cyberbullying involvement and its negative short-term psychosocial consequences appear to be well documented, further longitudinal research is needed to assess its longer-term impacts. The specific contents and perceived meanings of cyberbullying acts (e.g., text and photos posted on social media) may be elicited from a variety of stakeholders, including adolescents, parents, and teachers. Among cyberbullying perpetrators, their motivations and the mechanisms that may trigger or prevent acts of cyberbullying should be further investigated, as should resilience factors and coping mechanisms among victims. Given that same-aged peers may constitute the primary audience to events of cyberbullying, further research may also focus on adolescent bystanders and what might elicit helpful responses among them, such as supporting victims and speaking up against aggressive behaviors.

## Figures and Tables

**Figure 1 behavsci-14-01043-f001:**
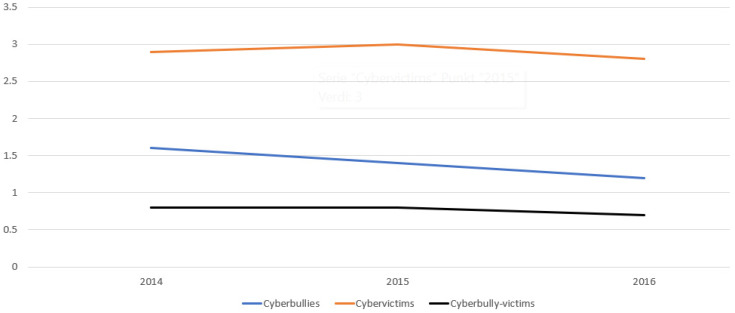
Cyberbullying perpetrators, victims, and cyberbully-victims 2014–2016 (%). *Note*. Participants classified as perpetrators and victims of cyberbullying, respectively, reported that they performed acts of cyber-/mobile phone bullying or were exposed to cyber-/mobile phone bullying at least once every 14 days. Cyberbully-victims reported performing acts of cyber-/mobile phone bullying and being exposed to cyber-/mobile phone bullying at least once every 14 days. In 2014, *n* varied between 44,590 and 44,747 for the three categories; in 2015, *n* varied between 70,621 and 70,846; and in 2016, *n* varied between 68,136 and 68,348.

**Table 1 behavsci-14-01043-t001:** Age and gender of the study participants within years.

Year	2014	2015	2016
Total sample *n*	46,019	73,426	70,577
*Age cohort*			
8th grade junior high	22.7%	19.8%	20.9%
9th grade junior high	22.9%	20.1%	20.2%
10th grade junior high	21.5%	19.7%	21.3%
1st grade senior high	15.0%	18.7%	21.6%
2nd grade senior high	10.5%	12.9%	12.9%
3rd grade senior high	7.4%	8.8%	3.2%
*Gender*			
Boys	50.6%	49.9%	50.5%
Girls	49.4%	50.1%	49.5%
*Socioeconomic status*			
Family affluence scale mean score (SD)	7.6 (1.3)	7.5 (1.3)	7.6 (1.3)

*Note*. Across the study period, there were 3.5% and 3.0% missing values on age and gender, respectively. Missing values for FAS varied between 2.0% and 6.9%.

**Table 2 behavsci-14-01043-t002:** Cyberbullying status cross tabulated with outcomes (data combined for 2014–2016).

	Belief in a good and happy life	
Cyberbullying status	No, or don’t know, *n* (%)	Yes, *n* (%)
Not perpetrator, not victim	42,858 (25.6)	124,408 (74.4)
Perpetrator, not victim	363 (39.5)	557 (60.5)
Victim, not perpetrator	1924 (54.4)	1616 (45.6)
Perpetrator and victim	564 (49.9)	567 (50.1)
	Loneliness	
Cyberbullying status	Not or little affected, *n* (%)	Quite or much affected, *n* (%)
Not perpetrator, not victim	129,175 (81.1)	30,127 (18.9)
Perpetrator, not victim	634 (71.7)	250 (28.3)
Victim, not perpetrator	1292 (37.4)	2163 (62.6)
Perpetrator and victim	659 (60.9)	423 (39.1)
	School thriving	
Cyberbullying status	No, *n* (%)	Yes, *n* (%)
Not perpetrator, not victim	9086 (5.9)	145,862 (94.1)
Perpetrator, not victim	242 (26.6)	669 (73.4)
Victim, not perpetrator	1143 (33.2)	2302 (66.8)
Perpetrator and victim	376 (31.9)	801 (68.1)
	Depressive symptoms	
Cyberbullying status	None or low level, *n* (%)	High level, *n* (%)
Not perpetrator, not victim	131,591 (83.9)	25,310 (16.1)
Perpetrator, not victim	616 (71.5)	246 (28.5)
Victim, not perpetrator	1462 (43.2)	1922 (56.8)
Perpetrator and victim	636 (60.8)	410 (39.2)

*Note*. For all group differences, *p* < 0.001 (Chi Squared test).

**Table 3 behavsci-14-01043-t003:** Logistic regression analyses displaying associations between cyberbullying status and outcomes, adjusted by age, gender, and friendship (data combined for 2014–2016).

Independent variables	Belief in a good and happy life (*n* = 162,487)		
	OR	95% CI	*p*
Age	0.93	0.92–0.93	<0.001
Gender	0.72	0.70–0.73	<0.001
At least one friend	2.25	2.20–2.31	<0.001
Cyberbullying status			
Not perpetrator, not victim	reference		
Perpetrator, not victim	0.49	0.43–0.57	<0.001
Victim, not perpetrator	0.31	0.29–0.34	<0.001
Perpetrator and victim	0.35	0.31–0.40	<0.001
	Loneliness (*n* = 154,813)		
	OR	95% CI	*p*
Age	1.17	1.16–1.18	<0.001
Gender	3.11	3.02–3.20	<0.001
At least one friend	0.40	0.39–0.41	<0.001
Cyberbullying status			
Not perpetrator, not victim	reference		
Perpetrator, not victim	2.22	1.89–2.60	<0.001
Victim, not perpetrator	7.15	6.62–7.73	<0.001
Perpetrator and victim	3.34	2.91–3.82	<0.001
	School thriving (*n* = 150,380)		
	OR	95% CI	*p*
Age	0.95	0.94–0.96	<0.001
Gender (male is reference)	0.77	0.74–0.80	<0.001
At least one friend	2.03	1.95–2.11	<0.001
Cyberbullying status			
Not perpetrator, not victim	Reference		
Perpetrator, not victim	0.17	0.14–0.19	<0.001
Victim, not perpetrator	0.14	0.13–0.15	<0.001
Perpetrator and victim	0.14	0.12–0.16	<0.001
	Depressive symptoms (*n* = 152,480)		
	OR	95% CI	*p*
Age	1.27	1.26–1.28	<0.001
Gender (male is reference)	4.21	4.08–4.35	<0.001
At least one friend	0.56	0.55–0.58	<0.001
Cyberbullying status			
Not perpetrator, not victim	Reference		
Perpetrator, not victim	3.32	2.82–3.90	<0.001
Victim, not perpetrator	7.44	6.88–8.03	<0.001
Perpetrator and victim	5.11	4.45–5.87	<0.001

*Note*. Reference values are male gender, not having a friend, and no involvement in cyberbullying neither as perpetrator nor as victim.

## Data Availability

The data analyzed in this study are subject to the following licenses/restrictions: The data and materials from the Ungdata Surveys are stored in a national database administered by NOVA. The data are available for research purposes upon application. Further information about the study and the questionnaires can be found on the web page https://www.nsd.no/nsddata/serier/ungdata_eng.html. Requests to access these datasets should be directed to ungdata@oslomet.no.

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
