# Peer review of "Cyberbullying Among Adolescents in Norway: Time Trends and Factors Associated with Perpetration and Victimization"

_behavsci, 2024, doi:10.3390/bs14111043_

Round 1
Reviewer 1 Report
Comments and Suggestions for Authors
Introduction:
1. The authors need to have a theoretical background about the development of cyberbullying and to justify more why adolescents are at higher risk of engaging in cyberbullying.
2. The different types of cyberbullying need to be clarified.
3. A more explicit link between cyberbullying, depression, and loneliness is needed.
4. The purpose of the study needs more coherent support. The author did not mention specific hypotheses.
Methods:
This section needs significant improvement.
5. The author should provide more detailed demographic information about the participants, including age, gender, and socio-economic status, to give a clearer picture of the study's sample. Additionally, it needs to provide more information about the sampling and exclusion/inclusion criteria.
6. The cross-sectional methodology is not appropriate to evaluate cyberbullying trends. If the authors collected longitudinal data, they must provide the attrition rates.
Results
This section is well written. However, authors need to:
7. Explain the statistical analysis plan better and provide further information about predictors and outcome variables.
8. Table 2 is suggested to present with a bar chart rather than a table to better understand the results.
Discussion:
This section needs significant improvement.
9. The findings needed to be better explained and linked with previous literature.
10. The information in the limitation section is not accurate. The data were selected from 2014-2016, but the authors mentioned different years of data collection.
11. There is a lack of methodological limitations.
12. Authors should provide clinical applications of their findings to strengthen the purpose of the current study.
Author Response
Authors: We thank the reviewers for their comments and suggestions, and we believe addressing them has improved the manuscript. All changes in the manuscript have been performed using track changes. We look forward to hearing from you again.
Reviewer 1 (R1): Introduction: The authors need to have a theoretical background about the development of cyberbullying and to justify more why adolescents are at higher risk of engaging in cyberbullying.
Authors: In the Introduction section (line 41-46 and line 95-100), we have included relevant materials, with appropriate references, about development and the higher risk among adolescents.
R1: The different types of cyberbullying need to be clarified.
Authors: Types of cyberbullying have been clarified in the Introduction section (line 54-55).
R1: A more explicit link between cyberbullying, depression, and loneliness is needed.
Authors: Cyberbullying and depression were explicitly linked in the first version of the manuscript, and we appreciate the reminder to also include loneliness among the factors related to cyberbullying (line 50).
R1: The purpose of the study needs more coherent support. The author did not mention specific hypotheses.
Authors: We appreciate the previous comments by the reviewer. We believe that responding to them with added materials in the Introduction section has provided sufficient support for the study. Hypotheses have been added (line 126-129).
R1: Methods: This section needs significant improvement. The author should provide more detailed demographic information about the participants, including age, gender, and socio-economic status, to give a clearer picture of the study's sample. Additionally, it needs to provide more information about the sampling and exclusion/inclusion criteria.
Authors: Class cohort was used as a proxy for age. Table 1 shows the age (i.e., class cohort) and gender proportions in the sample for each of the study years, and we have added the Family Affluence Scale mean score as a measure of socioeconomic status (see revised Table 1). Description of the measure is provided (line 198-207). More information about sampling and criteria for inclusion is provided (line 134-137).
R1: The cross-sectional methodology is not appropriate to evaluate cyberbullying trends. If the authors collected longitudinal data, they must provide the attrition rates.
Authors: While the repeated cross-sectional design is not optimal (a proper longitudinal design using the same participants at each time of assessment would be preferred, but was not an option with the currently available data), we believe it provides trustworthy prevalence estimates of cyberbullying involvement in each of the three study years, particularly in view of the very large sample size in each year and the nationwide implementation of the Ungdata surveys. The data are generally considered reliable and representative for Norwegian adolescents nationally.
R1: Results: This section is well written. However, authors need to: Explain the statistical analysis plan better and provide further information about predictors and outcome variables.
Authors: Without knowing what the reviewer is missing specifically, we have expanded the analysis description somewhat, first and foremost by including the motivation for the employed analyses. We also added that socioeconomic status was analyzed descriptively, and clarified that cyberbullying status was the predictor variable. Information about all employed variables is provided in the Methods/Measures section.
R1: Table 2 is suggested to present with a bar chart rather than a table to better understand the results.
Authors: Bar charts are generally appropriate for illustrating differences in magnitude of some sort. Table 2 displays 16 such differences (i.e., one for each row in the Table), and we believe presenting this information in the Figure format would be excessive. To understand the results, the reader should compare the proportions listed in each row and then consider the pattern of differences related to each of the outcomes. A summary of the results is also provided in the text (section 3.3).
R1: Discussion: This section needs significant improvement. The findings needed to be better explained and linked with previous literature.
Authors: We disagree. We believe all of the main results, both concerned with time trends and factors associated with cyberbullying, have been appropriately addressed and linked with previous literature. The discussion section contains no less than 20 references to foregoing studies.
R1: The information in the limitation section is not accurate. The data were selected from 2014-2016, but the authors mentioned different years of data collection.
Authors: Yes, the data used in this study is from 2014-2016. The other years mentioned in the limitations section concern the reasons for our inability to use more recent iterations of the Ungdata surveys.
R1: There is a lack of methodological limitations.
Authors: In addition to the already mentioned limitations, we have added limitations about self-reported measures (line 366-369), and that the associations found between cyberbullying status and outcomes might be influenced by other factors that were not controlled for in the present study.
R1: Authors should provide clinical applications of their findings to strengthen the purpose of the current study.
Authors: As the data come from a general population of adolescents, and not a clinical population (e.g., adolescents with mental illness), there can be no 'clinical implications' as such. We have added the most relevant practical implications of the study to the Conclusion section (line 382-394).
Reviewer 2 Report
Comments and Suggestions for Authors
In the contextualisation, it is also more than necessary to point out the differentiation in the use of mobile phones by pupils between the majority option (leisure, free time) and academic training (more residual). This is why delimiting the pragmatics of use is so important from the outset.
Are the demographic characteristics of the study related to factors such as COVID confinement, adolescent suicide rates and/or biorhythm issues based on the absence of daylight hours during the year?
It would be useful to indicate what legislation (if any) regulates the use of mobile devices and access to social networks as established by the European Union and most member countries.
On lines 280 to 291, it would be interesting to have the most recent data (even more so after the COVID pandemic and the technological development that has taken place over the last 4 years with social networks). Delimiting the data may cause information bias to the current reality.
In the conclusions (from line 351) it would be recommended to establish future lines of work that delve deeper into both cases of cyberbullying and mental health mediated by the use of mobile devices. Obtaining contemporary data would be an incentive to update the database on which the article is based.
Author Response
Authors: We thank the reviewers for their comments and suggestions, and we believe addressing them has improved the manuscript. All changes in the manuscript have been performed using track changes. We look forward to hearing from you again.
Reviewer 2 (R2): In the contextualisation, it is also more than necessary to point out the differentiation in the use of mobile phones by pupils between the majority option (leisure, free time) and academic training (more residual). This is why delimiting the pragmatics of use is so important from the outset.
Authors: We agree that it would have been interesting to investigate this difference, but this is not something we have data on, and it is not part of the aim of this article. However, we would argue that smartphones may be used for cyberbullying anywhere. Thus, using them e.g. in school does not necessarily preclude that they can be used for cyberbullying purposes. Perhaps more importantly, the consequences for victims would not be any different based on where a hurtful social media message was posted.
R2: Are the demographic characteristics of the study related to factors such as COVID confinement, adolescent suicide rates and/or biorhythm issues based on the absence of daylight hours during the year?
Authors: The present study used data collected a long time before COVID. We have no information about suicide rates, biorhythm issues or daylight hours.
R2: It would be useful to indicate what legislation (if any) regulates the use of mobile devices and access to social networks as established by the European Union and most member countries.
Authors: The purpose of the study was to examine time trends and factors associated with cyberbullying involvement in Norway. Legislation issues do not fall within our scope.
R2: On lines 280 to 291, it would be interesting to have the most recent data (even more so after the COVID pandemic and the technological development that has taken place over the last 4 years with social networks). Delimiting the data may cause information bias to the current reality.
Authors: We agree that having more recent data would be interesting, and that the available data from 2014-2016 might not be fully reflective of the current reality. However, we cannot use more recent data without it becoming incompatible with the data we have already used. This is stated both in the Discussion (line 309-311) and the Limitations sections (line 354-358).
R2: In the conclusions (from line 351) it would be recommended to establish future lines of work that delve deeper into both cases of cyberbullying and mental health mediated by the use of mobile devices. Obtaining contemporary data would be an incentive to update the database on which the article is based.
Authors: We appreciate the encouragement to provide examples of further venues for research in this field; see added text (line 395-407).